# DreamCast: Fast and Stable Text-to-3D Generation with Confidence-Driven Casting and Subject-Aligned Optimization

## Abstract

Text-to-3D generation has witnessed remarkable progress, largely driven by optimization-based approaches that offer significant potential for creative applications. Despite their promise, these methods are fundamentally constrained by high-variance diffusion priors, which frequently results in slow convergence and unstable geometry-texture generation. To address these limitations, we present DreamCast, a unified framework that achieves fast and stable text-to-3D generation with Confidence-Driven Casting and Subject-Aligned Optimization. Specifically, Confidence-Driven Casting fuses multi-view predictions and is established based on confidence to construct a robust, outlier-free geometric initialization. Subject-Aligned Optimization collapses the broad generative distribution onto a specific instance manifold, thereby preventing semantic drift. Furthermore, we introduce a continuous guidance annealing schedule that reconciles the alignment between explicit representations and diffusion gradients, ensuring a balanced trade-off between geometric convergence and texture fidelity during generation. Extensive experiments demonstrate that DreamCast achieves superior 3D content quality with greater stability and reduced generation time compared to existing methods.

## 1. Introduction

Text-to-3D generation serves as a critical bridge between linguistic creativity and three-dimensional visual content, holding transformative potential for applications such as game production, virtual reality, and industrial design (Li et al., 2023; Zhang et al., 2024b; Xiang et al., 2025; Li et al.). With the rapid maturation of diffusion models in the 2D image domain (Rombach et al., 2022), a series of pioneering works have demonstrated that pretrained image diffusion priors can be distilled into 3D representations through Score Distillation Sampling (SDS), giving rise to the modern paradigm of optimization-based text-to-3D generation (Poole et al., 2023; Lin et al., 2023; Chen et al., 2023).

Building upon this paradigm, a large body of optimization-based methods has explored how to improve generation quality and efficiency. On one hand, several methods aim to enhance visual fidelity by introducing more sophisticated optimization objectives or preference guidance, including variational score distillation and human preference learning (Wang et al., 2023b; Ye et al., 2024; Zhou et al., 2025). On the other hand, approaches such as DreamGaussian exploit the efficiency of 3D Gaussian Splatting (3DGS) to achieve orders-of-magnitude speedups in generation (Kerbl et al., 2023; Tang et al., 2024; Yi et al., 2024; Chen et al., 2024). Despite the substantial progress in optimization objective design and representational efficiency, optimization-based text-to-3D methods remain highly sensitive to initialization under fast generation settings, with the root cause lying in the lack of reliable geometric constraints. Specifically, without explicit 3D priors, relying purely on 2D diffusion guidance renders the joint inference of geometry and appearance largely ambiguous. This leads to a highly non-convex optimization landscape, in which accumulated noisy gradients often misguide the process into poor local minima. The result is typically characterized by unreasonable geometric artifacts and significant misalignment between texture and geometry (Poole et al., 2023; Wang et al., 2023a).

Recent works attempt to alleviate this issue by introducing alternative forms of guidance or regularization, such as classifier-based score distillation, noise-free or invariant score formulations, and deterministic sampling priors (Yu et al., 2023; Katzir et al., 2023; Zhuo et al., 2024; Wu et al., 2024b), or by initializing optimization with point clouds generated by methods such as Point-E or Shap-E (Nichol et al., 2022; Jun & Nichol, 2023). However, such priors are inherently sparse and low-resolution, often lacking precise semantic alignment. They struggle to support complex compositional prompts, leading to semantic drift or structural

---

[1]Anonymous Institution, Anonymous City, Anonymous Region, Anonymous Country. Correspondence to: Anonymous Author <anon.email@domain.com>.

Preliminary work. Under review by the International Conference on Machine Learning (ICML). Do not distribute.

collapse during subsequent optimization, as evidenced by our empirical analysis in Sec. 4 (see Fig. 4).

To address these challenges, we propose DreamCast, a unified framework for fast and stable text-to-3D generation. Specifically, a confidence-driven casting strategy is introduced in DreamCast by fusing multi-view predictions under deterministic constraints to construct a robust and outlier-free geometric initialization, providing a reliable structural anchor for subsequent optimization. Building upon this foundation, a subject-aligned optimization mechanism is further introduced in DreamCast by constraining the originally broad generative space onto a specific instance manifold, substantially reducing diffusion gradient variance and suppressing semantic drift. Meanwhile, we design a continuous guidance annealing schedule in DreamCast that emphasizes geometric convergence in the early stages of optimization and progressively enhances texture refinement in later stages. This design enables a smooth transition from geometric calibration to texture polishing within approximately 20 minutes, producing realistic details while preserving geometric consistency.

In summary, our main contributions are as follows:

- We systematically analyze the limitations of sparse geometric priors under complex text prompts, and propose a confidence-driven geometric casting approach that constructs a dense and robust geometric prior, effectively alleviating multi-view ambiguity.

- We introduce an optimization mechanism that combines manifold constraints with guidance annealing, transforming subject-driven adaptation into a stable optimization constraint that effectively suppresses semantic drift in SDS-based optimization.

- Extensive experiments demonstrate that DreamCast achieves competitive performance in terms of geometric consistency, texture quality, and text–semantic alignment while significantly reducing generation time (approximately 20 minutes), effectively balancing generation efficiency with optimization quality.

## 2. Related Work

**Score Distillation and Optimization-based Synthesis.** With the rapid advancement of text-to-image diffusion models (Rombach et al., 2022), text-to-3D generation has attracted increasing attention in recent years. Due to the limited scale and diversity of available 3D datasets compared to 2D (Deitke et al., 2023), pioneering works such as DreamFusion (Poole et al., 2023) and Score Jacobian Chaining (Wang et al., 2023a) explore distilling score information from pretrained 2D diffusion models to optimize 3D representations, typically parameterized as Neural Radiance

Fields (NeRFs) (Mildenhall et al., 2021). This score distillation paradigm has demonstrated promising results and inspired a wide range of follow-up methods that improve fidelity, diversity, and robustness through modified score formulations, advanced guidance strategies, or extended temporal modeling (Lin et al., 2023; Chen et al., 2023; Wang et al., 2023b; Yu et al., 2023; Katzir et al., 2023; Zhu et al., 2023; Chung et al., 2023; Bai et al., 2025; Bahmani et al., 2024).

Despite steady progress, a notable gap remains between 3D and 2D generation in terms of generation speed, robustness under complex prompts, and alignment with human preferences (Ye et al., 2024; Zhou et al., 2025). To improve efficiency, recent works replace NeRF-based optimization with more explicit representations. In particular, Dream-Gaussian (Tang et al., 2024) adopts 3D Gaussian Splatting (3DGS) (Kerbl et al., 2023) to achieve significant acceleration, while GaussianDreamer (Yi et al., 2024) further enhances 2D-to-3D prior transfer within the 3DGS framework. However, unlike continuous NeRF fields that exhibit an inductive bias toward smooth geometry, discrete Gaussian representations are highly sensitive to initialization. Existing methods often rely on sparse geometric priors (e.g., Point-E (Nichol et al., 2022)) or random initialization, which provide only coarse or noisy constraints for complex prompts, leading to fragmented geometry or suboptimal local minima (Chen et al., 2024). Related efforts also investigate alternative problem settings such as single-image reconstruction, text-driven NeRF synthesis, and instruction-based 3D editing (Liu et al., 2023; Zhang et al., 2024a; Haque et al., 2023; Zhou et al., 2024), which focus on controllability or editability rather than optimization stability. In this work, we further bridge the gap between fast generation and high-quality optimization by introducing reliable geometric anchoring and subject-aligned diffusion guidance on top of efficient 3D Gaussian representations, enabling both rapid convergence and stable high-fidelity text-to-3D generation.

**Subject-Driven and Identity-Preserving Synthesis.** Customizing generative models to preserve specific subject identities has achieved remarkable success in 2D, with methods such as DreamBooth (Ruiz et al., 2023) and Textual Inversion (Gal et al., 2022) enabling the synthesis of user-defined concepts. Extending this paradigm to 3D, recent approaches including DreamBooth3D (Raj et al., 2023), DreamCraft3D (Sun et al., 2024), and MV-Adapter (Huang et al., 2025) leverage fine-tuning or adapter-based mechanisms to reconstruct specific assets consistent with a given identity.

While these methods primarily target *personalization* by transferring external concepts into 3D, we repurpose subject-driven adaptation for a fundamentally different objective:

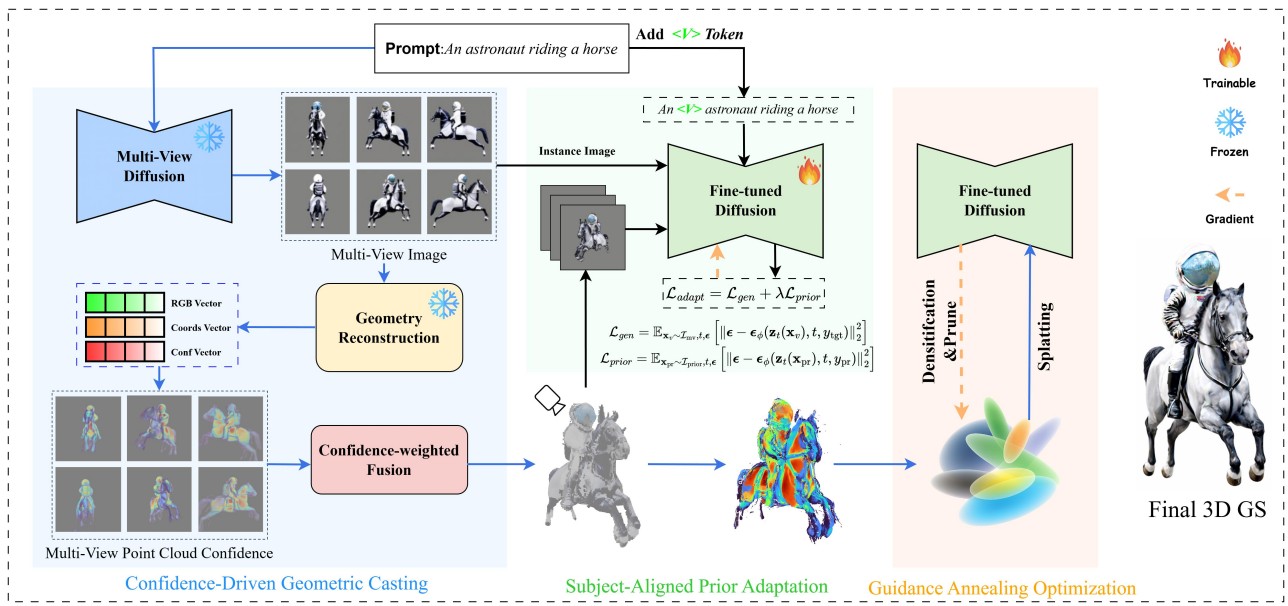

*Figure 1.* **Overview of the DreamCast framework.** The pipeline operates in three synchronized stages. **(Left) Confidence-Driven Casting:** Multi-view predictions from VGGT (Wang et al., 2025) are fused via confidence and density-aware downsampling to initialize robust 3D Gaussians. **(Middle) Subject-Aligned Prior:** We briefly adapt the diffusion model to the generated instance, narrowing the generative space to this specific subject to prevent semantic drift. **(Right) Unified Optimization:** The adapted prior drives stable SDS optimization under a guidance annealing schedule, ensuring high-fidelity geometry and texture convergence.

*optimization stability.* Standard SDS-based optimization is known to suffer from semantic drift (Metzer et al., 2023; Wang et al., 2023b), where the diffusion prior oscillates between multiple plausible interpretations of a generic text prompt. In contrast, DreamCast regards the coarse geometry derived from initialization as a pseudo-ground-truth subject. By dynamically anchoring the diffusion prior to this temporary identity, we effectively lock the optimization trajectory. This constraint collapses the multi-modal generative distribution onto a specific, consistent instance manifold, thereby mitigating semantic drift and the multi-faced "Janus" problem inherent in open-domain SDS optimization, without introducing any external identity or personalization signals.

## 3. Method

We present DreamCast, a unified framework bridging fast convergence and stable optimization. The complete pipeline is depicted in Fig. 1. We first review Score Distillation Sampling (Sec. 3.1), then introduce **Confidence-Driven Geometric Casting** for robust initialization (Sec. 3.2) and **Subject-Aligned Prior** to suppress semantic drift (Sec. 3.3). Finally, Sec. 3.4 describes a unified optimization scheme via continuous guidance annealing.

### 3.1. Preliminaries

**Score Distillation Sampling (SDS).** To generate 3D content from a text prompt $y$, we leverage a pre-trained text-

to-image diffusion model $\epsilon_\phi$. Since we lack ground-truth 3D data, optimization is driven by the Score Distillation Sampling (SDS) loss (Poole et al., 2023). SDS pushes the rendered image $\mathbf{x} = g(\Theta, v)$ at an arbitrary viewpoint $v$ towards high-probability regions of the diffusion prior. The gradient used to update $\Theta$ is defined as:

$$\nabla_\Theta \mathcal{L}_{\text{SDS}} = \mathbb{E}_{t,\epsilon} \left[ w(t) \left( \epsilon_\phi(\mathbf{z}_t; y, t) - \epsilon \right) \frac{\partial \mathbf{x}}{\partial \Theta} \right], \quad (1)$$

where $t \sim \mathcal{U}(0, 1)$ is the timestep, $\mathbf{z}_t$ is the noisy latent of the rendered image $\mathbf{x}$, $\epsilon$ is the added noise, and $w(t)$ is a weighting function. Standard SDS optimization typically initializes $\Theta$ randomly or within a bounded sphere, often leading to slow convergence and geometric artifacts due to the high variance of the diffusion prior $\epsilon_\phi$.

### 3.2. Confidence-Driven Geometric Casting

Optimization-based text-to-3D methods driven by SDS are highly sensitive to initialization. Random initialization often leads to slow convergence and multi-view artifacts, while recent 3DGS-based approaches relying on pretrained generators (e.g., Point-E or Shap-E) remain limited to simple prompts due to weak geometric reliability.

**Confidence-Driven Multi-View Reconstruction.** To remove this uncertainty, we construct a reliable geometric prior via dense multi-view reconstruction with explicit confidence filtering. Given $V$ multi-view images $\mathcal{I} =$

$\{I_1, \ldots, I_V\}$, we employ VGGT (Wang et al., 2025) under a cyclic reference permutation, producing an ensemble of point clouds $\{\mathcal{P}_k\}_{k=1}^V$. Each point $p$ is associated with a 3D position $\mathbf{x}_p$ and a confidence score $w_p$, reflecting the reliability of the reconstructed point. We aggregate all predictions into a voxel grid and compute, for each voxel $v$, a confidence-weighted centroid and aggregated confidence:

$$\bar{\mathbf{x}}_v = \frac{\sum_{p \in v} w_p \mathbf{x}_p}{\sum_{p \in v} w_p}, \quad W_v = \frac{1}{|v|} \sum_{p \in v} w_p. \quad (2)$$

Voxels with $W_v < \tau$ are pruned, retaining only geometry consistently supported across views, which effectively suppresses hallucinated or view-specific artifacts.

**Density-Aware 3DGS Initialization.** Directly instantiating Gaussians from high-confidence reconstructions often yields overly concentrated points, limiting flexibility during SDS refinement. We therefore normalize the fused point cloud to unit scale and perform density-aware voxel downsampling with resolution $\delta = 0.85\,d_{50}$, where $d_{50}$ is the median nearest-neighbor distance. After outlier removal, the remaining points are initialized as 3D Gaussians with scales proportional to $\delta$ and inherited colors, resulting in a compact and well-conditioned initialization. For detailed implementation, please refer to Appendix A.

### 3.3. Subject-Aligned Prior

Standard SDS optimizes a 3D representation by querying a frozen, open-domain diffusion prior $\boldsymbol{\epsilon}_\phi(\cdot|y)$. While effective for category-level generation, this paradigm becomes unstable when instance-level consistency is required. Under a generic prompt $y$, the conditional distribution $p_\phi(\mathbf{x}|y)$ is inherently multi-modal. Consequently, blind SDS optimization often causes the representation to oscillate among multiple plausible instances within the same semantic category, effectively ignoring the specific structural constraints provided by the initialization. We refer to this detrimental phenomenon as **Instance-Level Semantic Drift**, which fundamentally limits identity fidelity and corrupts the reliable geometric anchor.

To mitigate this issue, we propose a **Subject-Aligned Prior** that locally collapses the generative distribution onto a single-instance subspace. Instead of navigating the full open-domain manifold, we bias the optimization trajectory toward the specific subject defined by the initialization. Formally, we obtain the instance-aligned parameters $\phi$ by fine-tuning the pre-trained diffusion model to minimize the following objective:

$$\mathcal{L}_{\text{adapt}}(\phi) = \mathbb{E}_{\mathbf{x} \sim \mathcal{I}_{\text{mv}}, t, \boldsymbol{\epsilon}} \left[ \|\boldsymbol{\epsilon} - \boldsymbol{\epsilon}_\phi(\mathbf{z}_t(\mathbf{x}), t, y_{\text{tgt}})\|_2^2 \right] \quad (3)$$
$$+ \lambda \, \mathbb{E}_{\mathbf{x}_{\text{pr}} \sim \mathcal{I}_{\text{prior}}, t, \boldsymbol{\epsilon}} \left[ \|\boldsymbol{\epsilon} - \boldsymbol{\epsilon}_\phi(\mathbf{z}_t(\mathbf{x}_{\text{pr}}), t, y_{\text{pr}})\|_2^2 \right],$$

where $\mathcal{I}_{\text{mv}}$ denotes the set of multi-view consistent images generated by multi-view diffusion, and $y_{\text{tgt}}$ represents the target prompt augmented with a unique identifier token (e.g., "a $\langle V \rangle$ [class]") to bind the specific subject identity. The second term serves as a prior-preservation regularizer, controlled by the coefficient $\lambda$, designed to prevent catastrophic forgetting of the general diffusion prior. Here, $\mathcal{I}_{\text{prior}}$ comprises synthetic images generated by the frozen pre-trained model using the generic class prompt $y_{\text{pr}}$. From an optimization perspective, this adaptation constructs a local basin of attraction around the initialized subject within the score field, yielding directionally consistent SDS gradients. Consequently, semantic drift is effectively suppressed, ensuring that subsequent geometry and texture refinement reliably converge to the intended subject.

### 3.4. Unified Optimization via Joint Guidance Annealing

While the subject-aligned prior stabilizes semantic identity, SDS optimization still faces an inherent trade-off between coarse geometric convergence and fine-grained appearance fidelity.

**Guidance Strength Trade-off.** High classifier-free guidance scales are commonly employed in SDS optimization to encourage mode-seeking behavior, which is crucial for bootstrapping global geometry from the multi-modal diffusion prior (Chen et al., 2023; 2024). However, persistently high guidance is empirically observed to amplify gradient variance and over-emphasize high-contrast features, often leading to over-saturation and brittle textures in later stages. To address this issue, we decouple geometric bootstrapping from texture refinement along the optimization trajectory.

**Joint Annealing Schedule.** With a reliable geometric anchor (Sec. 3.2) and the subject-aligned prior (Sec. 3.3), prolonged high guidance becomes unnecessary. We therefore introduce a continuous joint annealing schedule that smoothly transitions the optimization focus from coarse geometry formation to fine-grained appearance refinement. Specifically, we linearly anneal the guidance scale $\omega(k)$ as:

$$\omega(k) = \max\left(\omega_{\min}, \, \omega_{\max} - \frac{k}{T_{\text{decay}}}(\omega_{\max} - \omega_{\min})\right), \quad (4)$$

where $k$ denotes the current iteration index, $\omega_{\max}$ and $\omega_{\min}$ represent the initial and final guidance scales respectively, and $T_{\text{decay}}$ defines the duration of the annealing phase. We find linear decay to be sufficient and stable in practice, without sensitivity to the exact decay form. In parallel, we progressively adjust the diffusion timestep range to emphasize structural alignment at early stages and appearance refinement at later stages. This joint schedule leverages strong structural gradients early on for point densification and pruning, while gradually relaxing the guidance to allow

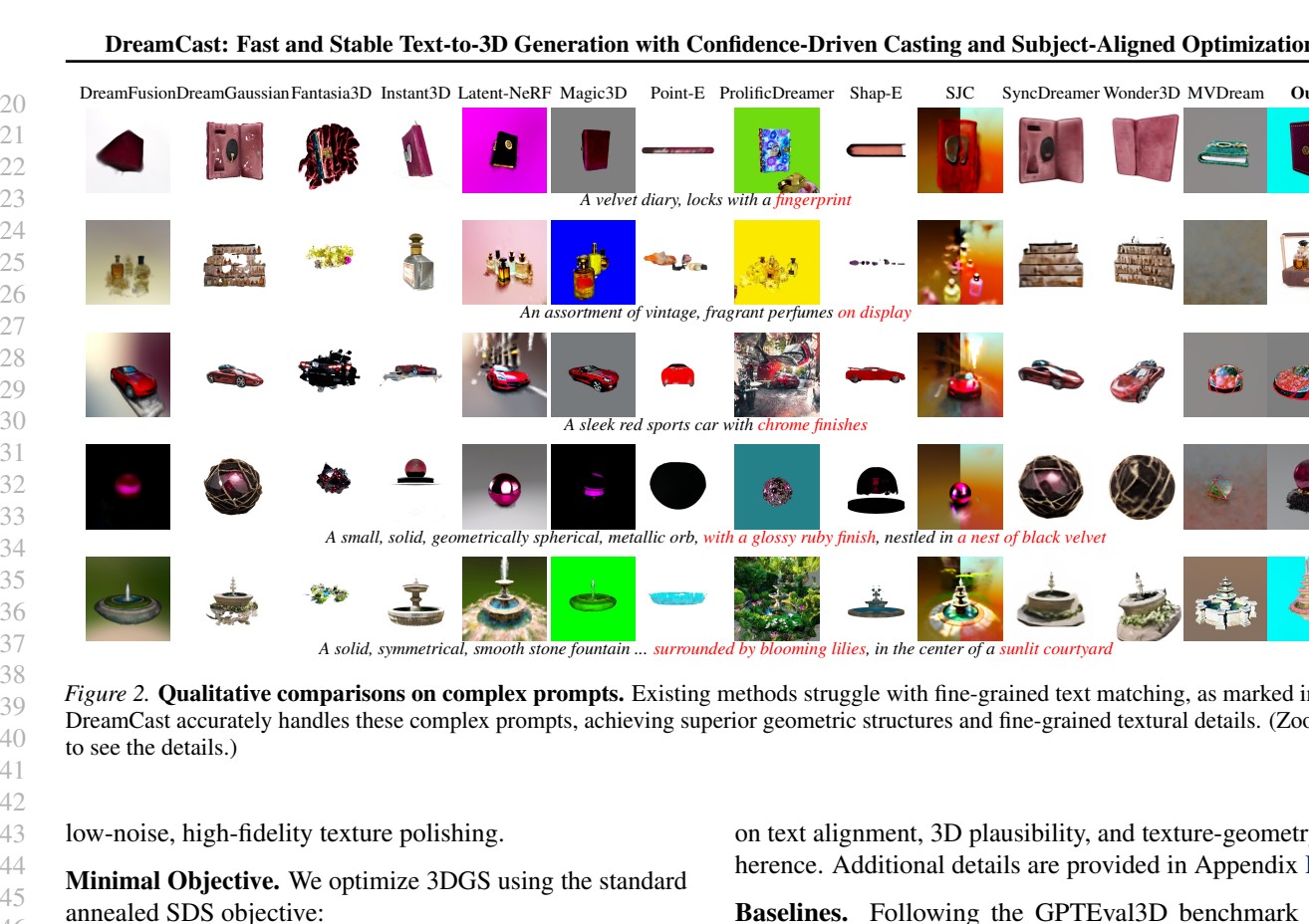

*Figure 2.* **Qualitative comparisons on complex prompts.** Existing methods struggle with fine-grained text matching, as marked in red. DreamCast accurately handles these complex prompts, achieving superior geometric structures and fine-grained textural details. (Zoom in to see the details.)

low-noise, high-fidelity texture polishing.

**Minimal Objective.** We optimize 3DGS using the standard annealed SDS objective:

$$\mathcal{L}_{\mathrm{SDS}}^{(k)} = \mathcal{L}_{\mathrm{SDS}}(\phi, \omega(k), t_{\max}(k)). \quad (5)$$

Unlike approaches that modify the loss formulation or gradient structure in SDS, our method stabilizes optimization by directly regulating the conditional diffusion guidance, effectively encouraging the optimization trajectory toward a single-instance solution while retaining the standard SDS objective.

## 4. Experiments

In this section, We evaluate DreamCast through a series of experiments. We first describe the experimental settings and implementation details (Sec. 4.1). We then compare our method with state-of-the-art approaches to demonstrate its effectiveness and efficiency (Sec. 4.2). Finally, we present ablation studies to analyze the contributions of key components in our framework (Sec. 4.3).

### 4.1. Experimental Setups

**Evaluation Protocol.** We evaluate DreamCast on 110 prompts from GPTEval3D (Wu et al., 2024a), covering diverse levels of creativity and compositional complexity. We use (i) ImageReward (Xu et al., 2023) scores averaged over 120 rendered views per asset, and (ii) GPT-4o pairwise comparisons against baselines, summarized by Elo ratings

on text alignment, 3D plausibility, and texture-geometry coherence. Additional details are provided in Appendix B.1.

**Baselines.** Following the GPTEval3D benchmark (Wu et al., 2024a), we primarily compare DreamCast with the 13 official baselines used in the benchmark, covering both text-guided and image-guided text-to-3D generation methods. The text-guided group includes DreamFusion (Poole et al., 2023), DreamGaussian (Tang et al., 2024), Instant3D (Li et al., 2024), Fantasia3D (Chen et al., 2023), Latent-NeRF (Metzer et al., 2023), Magic3D (Lin et al., 2023), MVDream (Shi et al., 2023), Point-E (Nichol et al., 2022), ProlificDreamer (Wang et al., 2023b), Shap-E (Jun & Nichol, 2023), and SJC (Wang et al., 2023a). The image-guided baselines include SyncDreamer (Liu et al., 2024) and Wonder3D (Long et al., 2024). In addition, we report results of two recent reward-guided methods, DreamReward (Ye et al., 2024) and DreamDPO (Zhou et al., 2025), as a complementary analysis to assess the effectiveness of DreamCast.

**Implementation Details.** We build on PyTorch (Paszke et al., 2019) and threestudio (Guo et al., 2023). Multi-view priors are synthesized using MV-Adapter (Huang et al., 2025), and geometry is reconstructed with VGGT (Wang et al., 2025). We apply a confidence threshold of 80 in the VGGT confidence space and voxel downsampling, resulting in approximately 100k points for 3DGS initialization. We adopt Stable Diffusion v2.1-base (Rombach et al., 2022) as the diffusion backbone and adaptation for 400 steps (about 5 minutes) with $\lambda = 1.0$ to align the diffusion prior with the initialized geometry. We then run 4,000 SDS iterations with

*Table 1.* **Quantitative comparisons on 110 prompts generated by GPTEval3D (Wu et al., 2024a).** The metric data (GPTEval3D scores) are provided by authors of (Wu et al., 2024a; Zhou et al., 2025). We calculate the ImageReward score (IR) (Xu et al., 2023) for human preference evaluation. The best performance is shown in **bold**, and the second best is underlined.

| Method | IR ↑ | GPTEval3D ↑ | | | | | |
|---|---|---|---|---|---|---|---|
| | | Alignment | Plausibility | T-G Coherency | Geo Details | Tex Details | Overall |
| DreamFusion (Poole et al., 2023) | -1.51 | 1000.0 | 1000.0 | 1000.0 | 1000.0 | 1000.0 | 1000.0 |
| DreamGaussian (Tang et al., 2024) | -1.56 | 1100.6 | 953.6 | 1158.6 | 1126.2 | 1130.8 | 951.4 |
| Fantasia3D (Chen et al., 2023) | -1.40 | 1067.9 | 891.9 | 1006.0 | 1109.3 | 1027.5 | 933.5 |
| Instant3D (Li et al., 2024) | -0.91 | 1200.0 | 1087.6 | 1152.7 | 1152.0 | 1181.3 | 1097.8 |
| Latent-NeRF (Metzer et al., 2023) | -0.42 | 1222.3 | 1144.8 | 1156.7 | 1180.5 | 1160.8 | 1178.7 |
| Magic3D (Lin et al., 2023) | -1.11 | 1152.3 | 1000.8 | 1084.4 | 1178.1 | 1084.6 | 961.7 |
| Point-E (Nichol et al., 2022) | -2.24 | 725.2 | 689.8 | 688.6 | 715.7 | 745.5 | 618.9 |
| ProlificDreamer (Wang et al., 2023b) | -0.50 | 1261.8 | 1058.7 | 1152.0 | 1246.4 | 1180.6 | 1012.5 |
| Shap-E (Jun & Nichol, 2023) | -2.10 | 842.8 | 842.4 | 846.0 | 784.4 | 862.9 | 843.8 |
| SJC (Wang et al., 2023a) | -0.82 | 1130.2 | 995.1 | 1033.5 | 1079.9 | 1042.5 | 993.8 |
| SyncDreamer (Liu et al., 2024) | -1.77 | 1041.2 | 968.8 | 1083.1 | 1064.2 | 1045.7 | 963.5 |
| Wonder3D (Long et al., 2024) | -1.70 | 985.9 | 941.4 | 931.8 | 973.1 | 967.8 | 970.9 |
| MVDream (Shi et al., 2023) | -0.58 | 1270.5 | 1147.5 | 1250.6 | 1324.9 | 1255.5 | 1097.7 |
| DreamReward (Ye et al., 2024) | -0.45 | 1287.5 | **1195.0** | 1254.4 | 1295.5 | 1261.6 | 1193.3 |
| DreamDPO (Zhou et al., 2025) | -0.35 | 1298.9 | 1171.9 | 1276.4 | **1373.2** | 1296.9 | 1203.1 |
| **DreamCast (Ours)** | **-0.24** | **1304.1** | 1182.4 | **1300.5** | 1279.5 | **1315.2** | **1304.9** |

*Table 2.* **Training time and representation comparison.** We compare DreamCast with prior text-to-3D methods in terms of training time (TT), representation (RP), and ImageReward (IR).

| Method | RP | TT | IR ↑ |
|---|---|---|---|
| DreamFusion (Poole et al., 2023) | NeRF | ∼1.5 hours | −1.51 |
| Magic3D (Lin et al., 2023) | NeRF+Mesh | ∼1 hour | −1.11 |
| Fantasia3D (Chen et al., 2023) | DMTet | ∼1 hour | −1.40 |
| Latent-NeRF (Metzer et al., 2023) | NeRF | ∼1 hour | −0.42 |
| ProlificDreamer (Wang et al., 2023b) | NeRF (VSD) | ∼8 hours | −0.50 |
| MVDream (Shi et al., 2023) | NeRF | ∼2 hours | −0.58 |
| DreamGaussian (Tang et al., 2024) | 3DGS+Mesh | ∼2 mins | −1.56 |
| GaussianDreamer (Yi et al., 2024) | 3DGS | ∼15 mins | −0.55 |
| **DreamCast (Ours)** | 3DGS | **∼20 mins** | **-0.24** |

guidance annealing from $\omega=100$ to 7.5 (until step 1,800), while annealing $t_{max}$ from 0.98 to 0.5. All experiments are conducted on a single NVIDIA A800 GPU, taking about **20 minutes** per prompt.

### 4.2. Comparison with Prior Methods

**Qualitative Comparisons.** We qualitatively evaluate DreamCast on challenging prompts that involve complex compositions and multiple interacting subjects. As shown in Figs. 2 and 3, many existing text-to-3D methods implicitly favor single-object generation and often struggle when multiple subjects are required by the prompt. In such cases, secondary objects are frequently missing, fused with the primary structure, or rendered with inconsistent geometry and appearance. In contrast, DreamCast robustly supports multi-subject synthesis and faithfully preserves the spatial relationships specified by complex prompts. Distinct object instances are consistently reconstructed with coherent geometry, while fine-grained material attributes and textures remain well aligned with the underlying structure. Compared to prior SDS-based approaches, our method exhibits improved texture fidelity and stronger texture–geometry consistency, avoiding the semantic confusion and instability commonly observed in compositional scenes. These results demonstrate that DreamCast generalizes beyond single-object synthesis and enables stable, high-quality text-to-3D generation under complex and compositional prompts. More qualitative results are provided in Appendix B.2 and B.3.

**Quantitative Comparisons.** We conduct quantitative evaluations on both performance and efficiency. Human preference is first evaluated using ImageReward (Tab. 1), where DreamCast achieves the highest score ($-0.24$), indicating stronger alignment with human aesthetic preferences. We further evaluate our method using GPTEval3D on 110 diverse text prompts. As shown in Tab. 1, DreamCast ranks first overall with an Elo score of 1304.9, substantially outperforming representative optimization-based baselines such as DreamGaussian and Fantasia3D. Our method remains highly competitive in 3D plausibility and geometric detail metrics, exhibiting balanced performance across all evaluation dimensions. Moreover, as shown in Tab. 2, DreamCast achieves relatively fast optimization while maintaining high visual quality, demonstrating a favorable trade-off among visual fidelity, geometric stability, and computational efficiency.

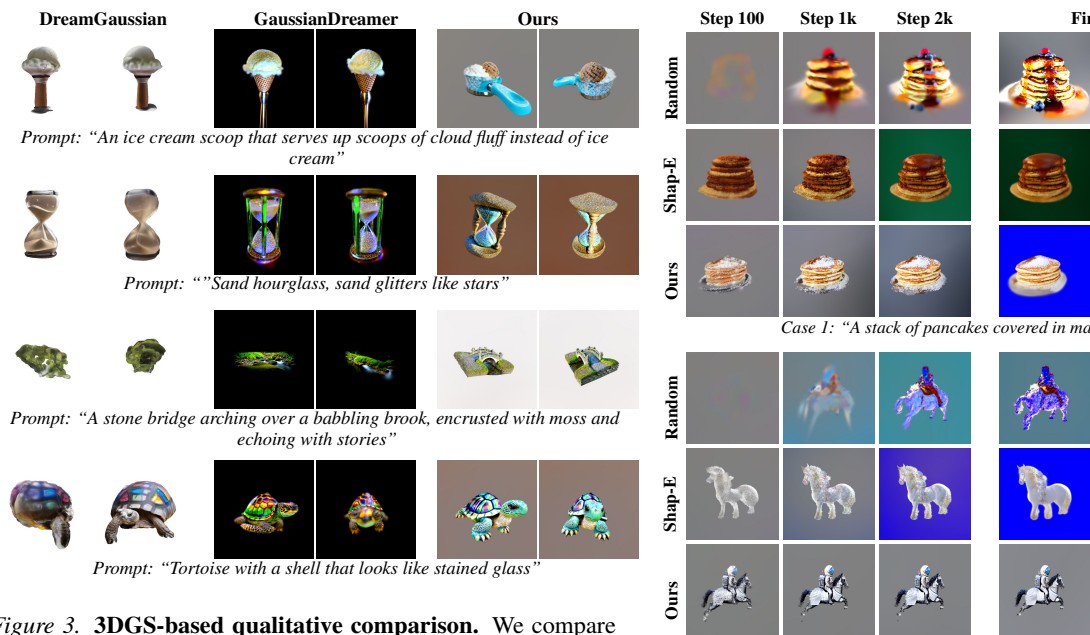

*Figure 3.* **3DGS-based qualitative comparison.** We compare DreamGaussian, GaussianDreamer, and our method on prompts from single object to complex scene. DreamCast produces more coherent geometry and better-aligned textures.

*Figure 4.* **Effect of initialization on convergence and stability.** For simple geometry (Case 1), all methods converge, whereas for complex compositions (Case 2), random initialization is unstable and Shap-E fails to recover multi-object structure. Our method consistently produces stable and high-fidelity results across runs.

*Table 3.* **Effect of guidance annealing on color saturation.** We report mean HSV saturation (Mean Sat.) and the oversaturation ratio (Oversat.) (foreground pixels with $S > 0.9$). Compared with constant high guidance, annealing from 100 to 7.5 reduces over-saturated regions.

| Setting | Mean Sat. | Oversat. ($S > 0.9$) |
|---|---|---|
| Constant (100) | $0.6916 \pm 0.0735$ | $0.3830 \pm 0.1304$ |
| Annealed (100→7.5) | $\mathbf{0.6619 \pm 0.1163}$ | $\mathbf{0.3018 \pm 0.1532}$ |

### 4.3. Ablation Study

**Initialization Strategies.** We study the impact of different initialization strategies on optimization stability and convergence behavior. Specifically, we compare our confidence-driven geometric initialization with random initialization and Shap-E initialization. As shown in Fig. 4, for simple object prompts (Case 1), all methods eventually converge, but our initialization achieves high-quality textures significantly faster, indicating smoother optimization. In contrast, under compositional prompts(Case 2), random initialization leads to severe instability and polyhedral artifacts, while Shap-E struggles to disentangle multi-object topology, resulting in collapsed or blurred geometry. Benefiting from dense and confidence-weighted geometric priors, our initialization consistently converges to semantically correct and structurally coherent geometry, preserving proper spatial relationships between multiple objects. These results suggest that strong geometric anchoring is crucial for stabilizing SDS-based optimization, especially under complex and compositional text prompts.

**Effect of the Subject-Aligned Prior.** We analyze the effect of the proposed subject-aligned prior on optimization stability and instance-level consistency. As shown in Fig. 5, removing the subject-aligned constraint causes the optimization to explore different visual modes within the same semantic category, leading to instance-level identity drift across iterations. Although the resulting geometry remains plausible, noticeable appearance inconsistencies emerge in texture and style. In contrast, incorporating the subject-aligned prior anchors diffusion guidance to the initialized geometry, enabling consistent recovery of the same instance across views and iterations. These results indicate that the subject-aligned prior effectively stabilizes SDS-based optimization by preventing unintended mode switching.

**Effect of Guidance Annealing.** We study the effect of guidance annealing on visual quality, with a focus on color saturation artifacts in SDS-based optimization. As shown in Tab. 3, maintaining a high guidance scale throughout training leads to over-saturated colors and amplified high-frequency artifacts. In contrast, guidance annealing significantly reduces both the mean HSV saturation and the proportion of over-saturated foreground pixels. We further observe that jointly annealing the maximum diffusion timestep $t_{\max}$ reinforces this effect by emphasizing low-noise diffusion steps that are more suitable for texture and material refinement. Qualitatively, Fig. 6 shows that guidance annealing suppresses unnatural color spikes while preserving fine-

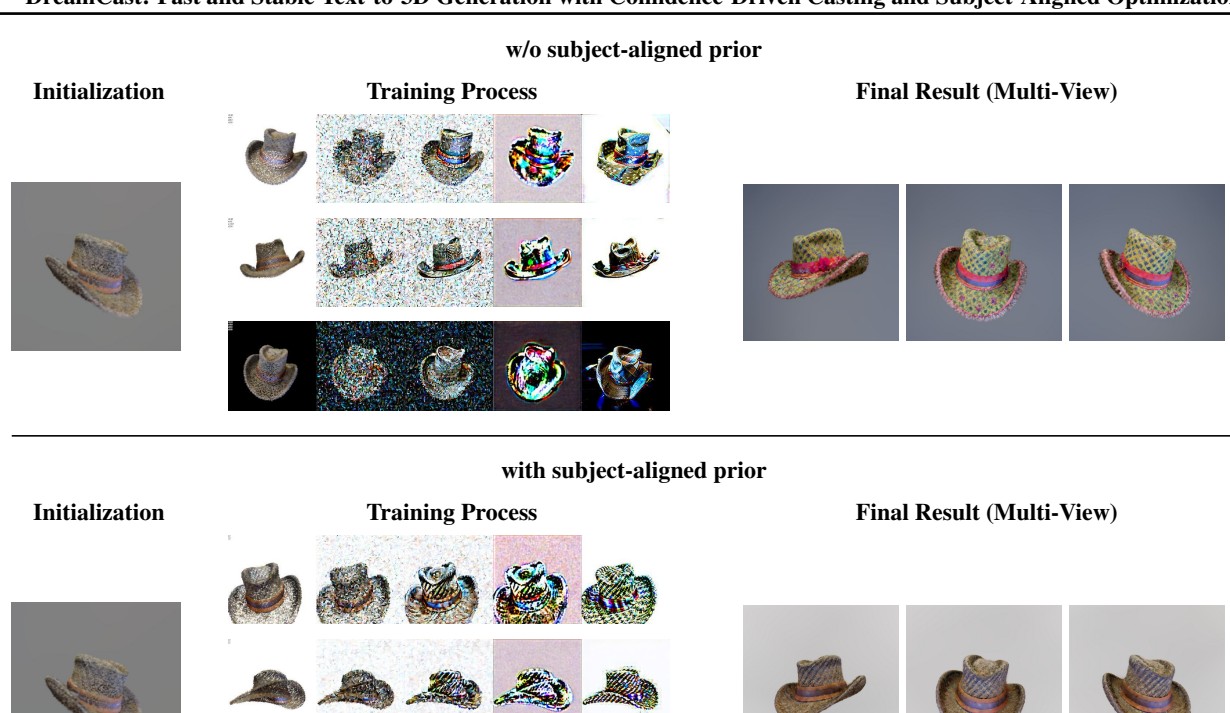

*Figure 5.* **Analysis of subject-aligned prior. Top (w/o subject-aligned prior):** The baseline suffers from **identity drift**, although it converges to a plausible 3D shape, the optimization explores different instances within the same category (e.g., various hats), leading to noticeable deviation from the input reference. **Bottom (Ours):** With the Subject-aligned Prior, the optimization remains anchored to the initialized subject, faithfully preserving the reference's material and texture details in the converged 3D asset.

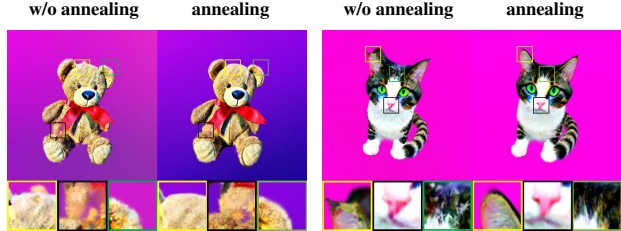

*Figure 6.* **Effect of guidance annealing on visual quality.** Visual comparison between constant high guidance (**left**) and the proposed annealing schedule (**right**). Guidance annealing suppresses over-saturation artifacts (highlighted) while preserving fine-grained texture details.

grained texture details. These results indicate that guidance annealing enables a stable coarse-to-fine optimization process, complementing geometric initialization and subject alignment for high-quality text-to-3D generation.

## 5. Conclusion

We presented DreamCast, a text-to-3D generation framework that balances optimization stability, visual fidelity, and efficiency. By combining confidence-driven geometric casting, a subject-aligned prior, and joint guidance annealing, DreamCast enables a stable coarse-to-fine SDS optimization process within a single loop. Experiments demonstrate strong geometric consistency, improved text alignment, and a favorable quality–efficiency trade-off compared to prior methods, achieving high-fidelity results within an optimization budget of approximately 20 minutes.

Limitations include reliance on the quality of geometric initialization and a manually designed guidance annealing schedule. Future work will explore adaptive guidance strategies, richer reward signals, and extensions to more complex or dynamic scenes.

## Impact Statement

DreamCast advances text-to-3D generation by providing a fast and stable optimization framework that can lower the barrier to producing coherent 3D assets. By improving the reliability and efficiency of score distillation-based optimization, our approach may benefit applications in game production, virtual and augmented reality, and industrial design, where flexible 3D content creation is desirable. From

an ethical perspective, our method inherits the limitations and biases of the pretrained 2D diffusion priors on which it relies, which can manifest in the generated 3D content. In addition, as with other generative models, there is a risk of misuse for producing misleading or unauthorized 3D assets, as well as an environmental cost associated with optimization-based pipelines. DreamCast focuses on improving optimization behavior rather than expanding generative capacity or introducing new training data. We plan to release our implementation to support transparency and further scrutiny, and we encourage practitioners to adopt responsible data curation, documentation, and deployment practices when applying text-to-3D generation systems.

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

## A. Geometric Initialization Algorithm

We provide a detailed description of our confidence-driven geometric casting strategy. Given a text prompt $y$, the algorithm constructs a dense and reliable point cloud that serves as the initialization for 3D Gaussian Splatting (3DGS).

---

**Algorithm 1** Confidence-Driven Geometric Casting

---

**Require:** Text prompt $y$, number of views $V$, confidence threshold $\tau_{\text{conf}}$
**Ensure:** Initialized 3D Gaussians $\Theta_0$
1: **Multi-view generation:**
2: Generate multi-view images $\{I_1, \ldots, I_V\}$ conditioned on $y$
3: **Permutation-based reconstruction:**
4: Initialize $\mathcal{P}_{\text{raw}} = \emptyset$
5: **for** $k = 1$ to $V$ **do**
6:     Construct cyclic view sequence $\mathcal{S}_k$
7:     Reconstruct geometry $\mathcal{P}_k = \{(\mathbf{x}_p, \mathbf{c}_p, w_p)\}$ using VGGT
8:     $\mathcal{P}_{\text{raw}} \leftarrow \mathcal{P}_{\text{raw}} \cup \mathcal{P}_k$
9: **end for**
10: **Confidence-aware fusion:**
11: Voxelize $\mathcal{P}_{\text{raw}}$ into grid $\mathcal{V}$
12: **for** each voxel $v \in \mathcal{V}$ **do**
13:     Collect points $\mathcal{S}_v$
14:     **if** $|\mathcal{S}_v| > 0$ and $\bar{w}_v \geq \tau_{\text{conf}}$ **then**
15:        Compute confidence-weighted centroid $(\bar{\mathbf{x}}_v, \bar{\mathbf{c}}_v)$
16:        Add fused point to $\mathcal{P}_{\text{fused}}$
17:     **end if**
18: **end for**
19: **Post-processing:**
20: Normalize $\mathcal{P}_{\text{fused}}$, remove outliers, obtain $\mathcal{P}_{\text{final}}$
21: **3DGS initialization:**
22: **for** each point $\mathbf{x}_i \in \mathcal{P}_{\text{final}}$ **do**
23:     Initialize Gaussian $g_i = (\boldsymbol{\mu}_i, \mathbf{s}_i, \mathbf{q}_i, \alpha_i, \mathbf{c}_i)$
24: **end for**
25: **return** $\Theta_0 = \{g_i\}$

---

## B. Supplementary Experimental Settings

### B.1. Details of Measurement Metrics

In the main paper, we employ two complementary evaluation strategies to demonstrate the superiority of the proposed DreamCast framework. Here, we provide supplementary details regarding these measurements.

**Evaluation with ImageReward.** ImageReward (Xu et al., 2023) is a text-to-image human preference reward model trained on large-scale human feedback data. Due to its high correlation with human aesthetics and prompt alignment, it has been broadly adopted for evaluation in both text-to-image and text-to-3D generation tasks (Ye et al., 2024; Zhou et al., 2025). Given a text prompt and a rendered image, ImageReward extracts multi-modal features and outputs a scalar score representing the preference confidence. To evaluate a 3D asset, we uniformly render 120 RGB images from diverse azimuths and elevations surrounding the object. The final ImageReward score for each asset is calculated by averaging the scores of these multi-view renderings.

**Evaluation with GPTEval3D.** We utilize GPTEval3D (Wu et al., 2024a), a comprehensive benchmark designed for large, multi-modal model-based evaluation of text-to-3D generation. The benchmark consists of theoretical comparison baselines, diverse prompts, and five distinct evaluation criteria: *text-asset alignment*, *3D plausibility*, *texture details*, *geometry details*, and *texture-geometry coherency*. Following the official protocol, we employ GPT-4o as the evaluator to conduct pairwise comparisons between 3D assets generated by our method and those from baseline methods under the same text prompt. These pairwise comparison results are aggregated to calculate the Elo rating for each model. Formally, let $\mathbf{A}$ be a matrix

where $A_{ij}$ represents the number of times the $i$-th model is preferred over the $j$-th model. The Elo ratings are derived by optimizing the following objective:

$$\sigma = \arg\max_{\sigma} \sum_{i \neq j} A_{ij} \log\left(\frac{1}{1 + 10^{(\sigma_j - \sigma_i)/400}}\right), \tag{6}$$

where $\sigma_i \in \mathbb{R}$ denotes the Elo rating of the $i$-th model. Consistent with standard practices, we calculate the Elo ratings within the existing tournament context, initializing and freezing the baseline scores as specified in the official GPTEval3D implementation (Wu et al., 2024a).

**Oversaturation Metric.** To quantify color oversaturation, we measure saturation in the HSV color space. Given a rendered RGB image, we convert it to HSV and extract the saturation channel $S \in [0, 1]$. The oversaturation ratio is defined as

$$r_{\text{over}} = \frac{1}{|\Omega|} \sum_{p \in \Omega} \mathbb{I}(S_p > 0.9), \tag{7}$$

where $\Omega$ denotes the set of foreground pixels and $\mathbb{I}(\cdot)$ is the indicator function.

To evaluate a 3D asset, we uniformly render 120 RGB images from diverse azimuths and elevations, and report the final oversaturation ratio by averaging the measurements over all views.

### B.2. GPTEval3D Evaluation Results

We provide visualizations of the GPTEval3D evaluation results across different dimensions. The following figures illustrate the comprehensive evaluation performance of various text-to-3D generation methods.

### B.3. Additional Qualitative Results

We provide additional qualitative comparisons between GaussianDreamer and our method across diverse prompts. The following figures demonstrate the consistent superiority of our approach in generating high-fidelity 3D assets with accurate text alignment and geometric details.

**"a plush teddy bear sitting alone with a slight tear"**

Ours      Magic3D

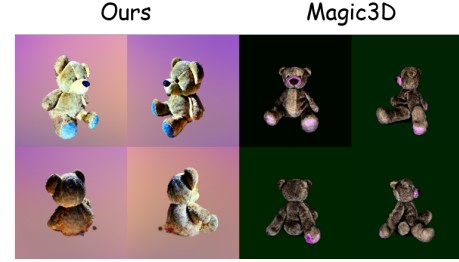

**Text prompt & Asset Alignment:** The left object presents a more traditional plush teddy bear look, capturing the essence of a teddy bear with a slight tear in its seam. The right object also depicts a teddy bear but with texture and details that differ slightly from what one might expect. The left one is better.
**3D Plausibility:** The left model appears solid with realistic proportions, resembling an actual teddy bear closely. The right model displays some irregularities, such as parts appearing less cohesive, making it less plausible. The left one is better.
**Geometry-Texture Alignment:** The left teddy bear has textures that align well with its geometry, maintaining a plush appearance. The right one, while still aligned, has some inconsistencies that detract from its overall appearance. The left one is better.
**Low-Level Texture Details:** The left object captures fine details well, enhancing its cuddly look without any blurriness. The right object, however, shows some abstractness in its texture, which may not align with the desired aesthetic. The left one is better.
**Low-Level Geometry Details:** The normal map of the left bear accurately reflects the plush texture and shape, providing a smooth and organized structure. The right bear, in contrast, has more noise and less intricate details, making it less visually pleasing. The left one is better.
**Overall:** the left model consistently outperforms the right across all criteria, demonstrating better alignment, plausibility, geometry-texture coherence, and detail work. The left one is better.
**Final answer:** left, left, left, left, left, left

---

**"An embroidered silk pillow with tassels"**

Ours      latent-nerf

**Text prompt & Asset Alignment:** The left object displays a colorful embroidered design with tassels, reflecting the prompt accurately; the right object has a simpler design and fewer details, missing some critical aspects of the prompt. The left one is better.
**3D Plausibility:** The left object's multi-view images show a more solid representation of a pillow with realistic proportions and details; the right object appears flatter and less plausible as a 3D item. The left one is better.
**Geometry-Texture Alignment:** The left model exhibits a consistent alignment between its textures and geometrical forms, while the right one has a mismatch, particularly in its surface design elements. The left one is better.
**Low-Level Texture Details:** The left object captures intricate embroidery details effectively, making it visually appealing and in alignment with the desired aesthetic. The right object's details are less defined and somewhat blurry. The left one is better.
**Low-Level Geometry Details:** The left object's normal map shows refined geometry that accurately reflects the desired pillow shape, while the right model displays less clarity and precision in its structural details. The left one is better.
**Overall:** the left 3D object consistently outperforms the right one across all criteria assessed. The left one is better.

**Final answer:** left, left, left, left, left, left

---

**"A dancing elephant"**

Ours      fantasia3d

**Text prompt & Asset Alignment:** The left model depicts the dancing elephant concept more vibrantly, with clear attributes such as the body posture and ear positions. The right model, while it does represent an elephant, lacks the dynamic impression of dancing. The left one is better.
**3D Plausibility:** The left model exhibits a more natural form and solid appearance, showcasing a well-defined shape that aligns closely with the characteristics of a realistic elephant. The right model appears more abstract and less believable in form. The left one is better.
**Geometry-Texture Alignment:** The texture of the left model adheres nicely to its geometry, giving a cohesive look. The right model's texture appears less aligned, with some irregularities that detract from the overall form. The left one is better.
**Low-Level Texture Details:** The left model captures fine details like skin texture and features without blurring, creating a realistic effect. The right model lacks the same level of detail and appears more simplistic and less engaging. The left one is better.
**Low-Level Geometry Details:** The normal map of the left model shows intricate details and smooth structure, representing the intended shape effectively. The right model shows signs of noise and less precision in its geometry. The left one is better.
**Overall:** the left model consistently outperforms the right model across all criteria in terms of realism, detail, and alignment with the prompt. The left one is better.
**Final answer:** left, left, left, left, left, left

---

**"Spotted ladybug crawling on a green leaf"**

Ours      dreamgaussian

**Text prompt & Asset Alignment:** The left model accurately depicts a spotted ladybug on green leaves, adhering closely to the text prompt. The right model fails to represent recognizable elements, appearing abstract and lacking defined shapes. The left one is better.
**3D Plausibility:** The left model shows a natural and coherent structure, while the right model appears fragmented and less solid, lacking the natural flow of an actual ladybug and leaf. The left one is better.
**Geometry-Texture Alignment:** The left model effectively aligns the texture of the ladybug and leaves with their geometry. In contrast, the right model lacks clear geometry and texture coherence. The left one is better.
**Low-Level Texture Details:** The left model captures fine details such as the ladybug's spots and the leaf's texture, maintaining clarity. The right model is overly abstract and lacks fine details, appearing blurry. The left one is better.
**Low-Level Geometry Details:** The left model demonstrates well-organized geometry that resembles real-world objects. The right model displays chaotic structures with irregular shapes that detract from its appeal. The left one is better.
**Overall:** the left model outperforms the right in all criteria, presenting a more accurate and visually appealing representation of the prompt.
The left one is better.
**Final answer:** left, left, left, left, left, left

*Figure 7.* **Results from GPTEval3D (Wu et al., 2024a).** We display the detailed pairwise comparison logs generated by GPT-4o. Detailed logs highlight specific strengths (green) and weaknesses (red).

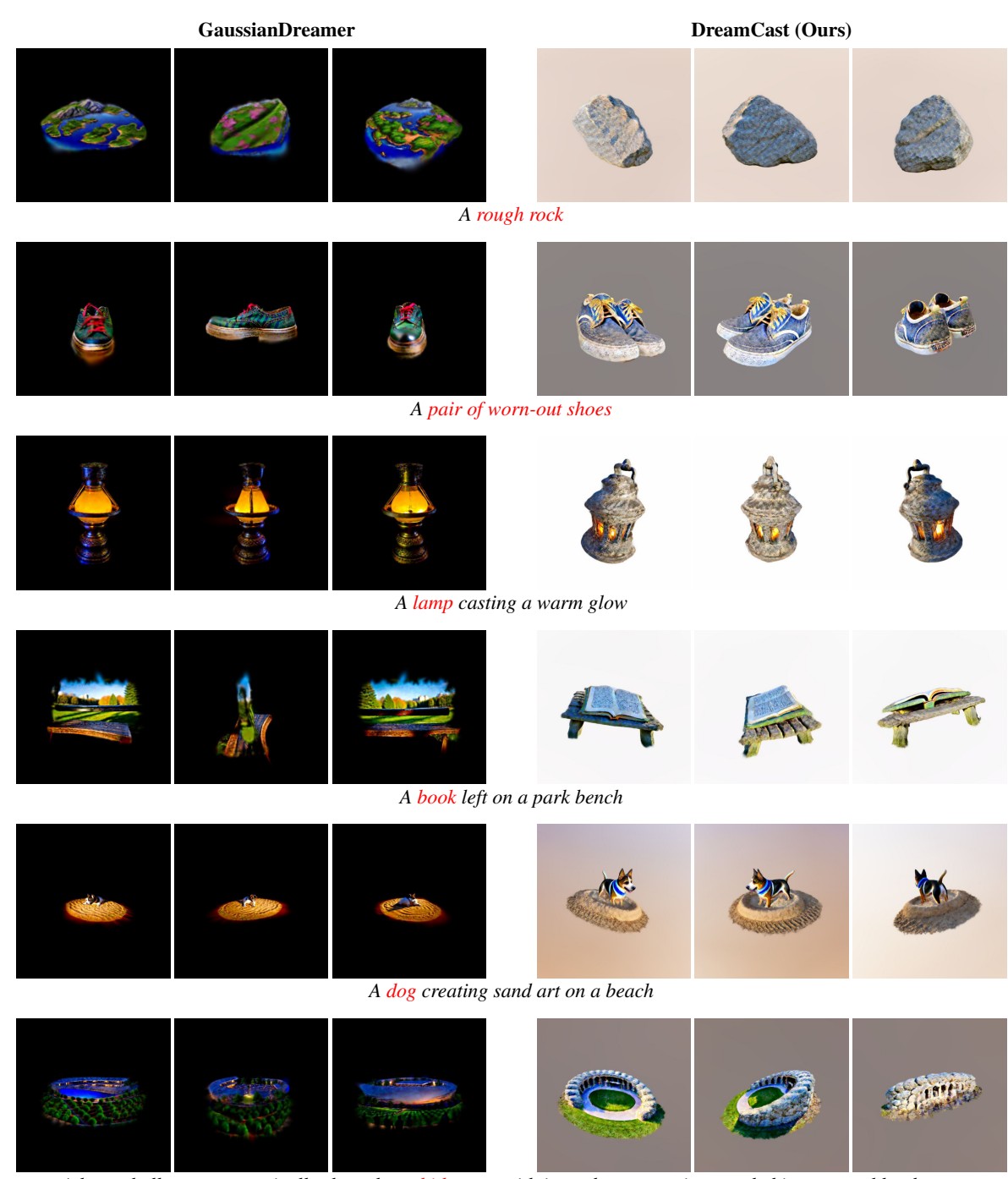

**GaussianDreamer**          **DreamCast (Ours)**

*A rough rock*

*A pair of worn-out shoes*

*A lamp casting a warm glow*

*A book left on a park bench*

*A dog creating sand art on a beach*

*A large, hollow, asymmetrically shaped amphitheater, with jagged stone seating, nestled in a natural landscape*

*Figure 8.* **Additional qualitative comparisons (Part 1).** Comparison between GaussianDreamer and DreamCast (Ours) on diverse prompts. Each row shows three views of the generated 3D assets.

**GaussianDreamer**             **DreamCast (Ours)**

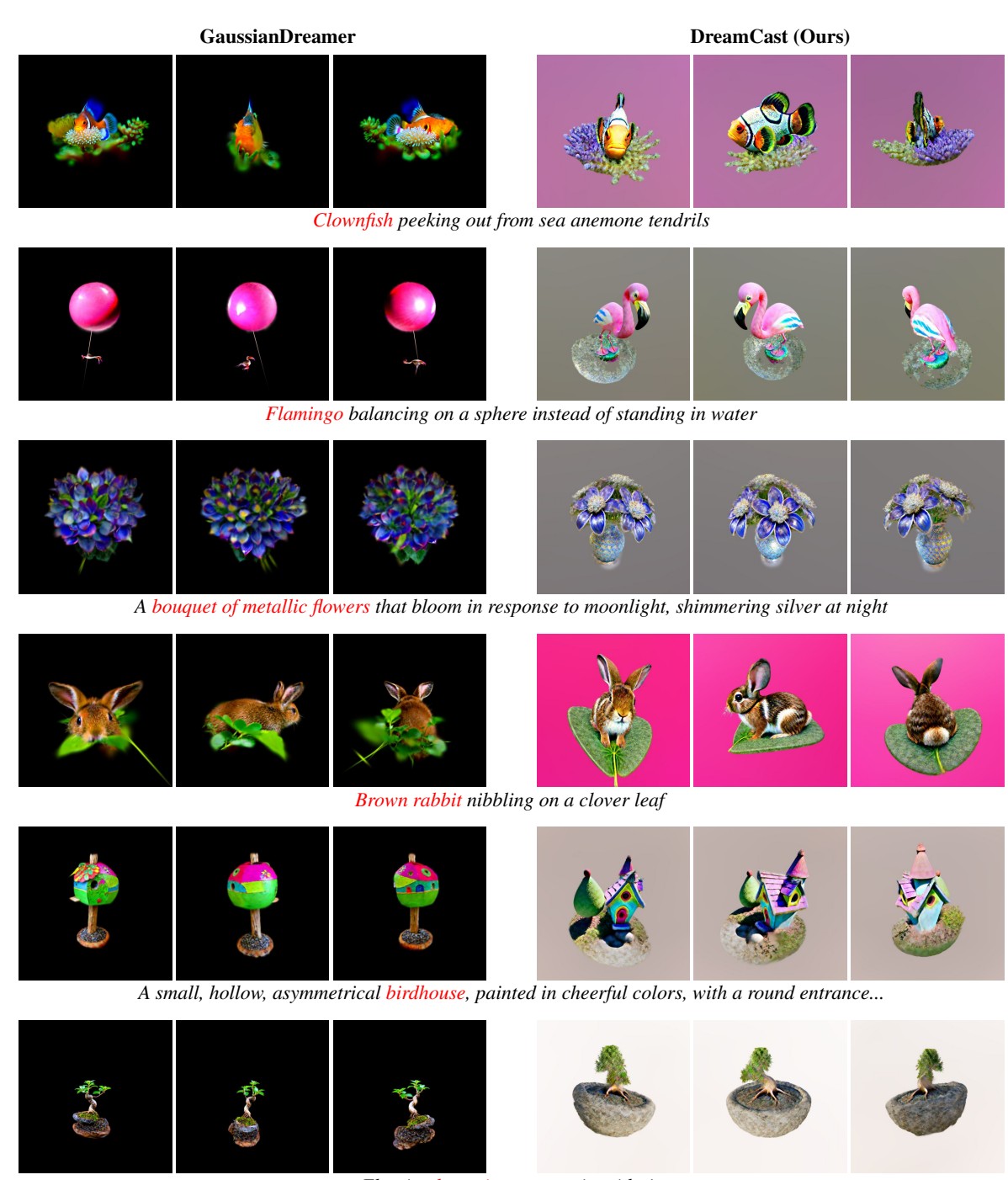

*Clownfish peeking out from sea anemone tendrils*

*Flamingo balancing on a sphere instead of standing in water*

*A bouquet of metallic flowers that bloom in response to moonlight, shimmering silver at night*

*Brown rabbit nibbling on a clover leaf*

*A small, hollow, asymmetrical birdhouse, painted in cheerful colors, with a round entrance...*

*Floating bonsai tree, roots in mid-air*

*Figure 9.* **Additional qualitative comparisons (Part 2).** Comparison between GaussianDreamer and DreamCast (Ours) on diverse prompts. Each row shows three views of the generated 3D assets.

