# OpenReview forum: "DreamCast: Fast and Stable Text-to-3D Generation with Confidence-Driven Casting and Subject-Aligned Optimization"
_ICML.cc/2026/Conference — Submitted to ICML 2026_

### Official Review · Reviewer_i9Dt · 2026-03-11

**Soundness:** 2
**Presentation:** 3
**Significance:** 2
**Originality:** 2
**Overall Recommendation:** 2
**Confidence:** 4

**Summary:**

## Summary

This paper presents DreamCast, an optimization-based text-to-3D pipeline that combines multi-view image synthesis, geometry reconstruction, short subject-specific diffusion adaptation, and annealed SDS optimization. The stated goal is to improve both stability and efficiency, so that higher-quality 3D assets can be produced in about 20 minutes.

**Compliance With Llm Reviewing Policy:**

Affirmed.

**Final Justification:**

I appreciate the authors’ rebuttal and the effort to clarify the paper’s motivation and positioning. However, my overall assessment remains unchanged. My main concern is still that the paper feels more like a stack of existing modules and a strong engineering pipeline than a clear methodological advance. I am also not convinced by the comparison setup: if the task goal is the same, stronger recent pipelines such as TRELLIS.2 are highly relevant, yet the paper mainly compares against weaker or older baselines, while still showing a noticeable qualitative gap to stronger systems. More broadly, I do not find the “optimization-based SDS” distinction sufficient to justify this choice, since methods like TRELLIS.2 also support text input, and the claimed flexibility still relies on large pretrained diffusion models. For these reasons, the rebuttal does not change my view, and I maintain my original score and recommendation.

**Key Questions For Authors:**

See weakness

**Limitations:**

Yes

**Strengths And Weaknesses:**

## Strengths

The paper studies a real problem. Optimization-based text-to-3D methods are often slow and unstable, especially when initialization is weak and the diffusion prior pulls the optimization toward inconsistent local solutions. In that context, the paper makes a reasonable case that better initialization and stronger optimization control can improve stability.

## Weaknesses

The main weakness is originality. The paper does not introduce a new optimization principle, a new distillation objective, or a clearly new 3D generation mechanism. Its main components are individually familiar: confidence-based fusion for initialization, short subject adaptation of a diffusion prior, and a simple annealing schedule for guidance. Since the method explicitly keeps the standard annealed SDS objective and does not change the core gradient formulation, the contribution reads more like a careful system integration paper than a real methodological advance.

The method design is also less compelling than the framing suggests. Much of the apparent gain may come from using strong upstream components such as MV-Adapter and VGGT, rather than from the proposed optimization strategy itself. The current draft does not isolate these sources of improvement clearly enough. As a result, it remains unclear whether DreamCast is a new method in its own right, or mainly a better initialization pipeline followed by standard SDS with mild scheduling changes.

The empirical comparison is not current enough. Most baselines come from the GPTEval3D benchmark and are largely 2023 to 2024 methods, while only very limited comparison is made to recent 2025 work. This is especially problematic because the proposed system itself relies on 2025 components. The comparison therefore feels asymmetric: the method benefits from a newer toolchain, but the main table is still dominated by older baselines. That makes the state-of-the-art claim much less convincing.

A closely related problem is the absence of comparison to image-to-3D or text-to-image-then-3D pipelines (eg. TRELLIS.2). Once the method already starts by generating multi-view images and reconstructing geometry from them, it is no longer enough to compare only against classic text-to-3D baselines. A natural question is whether a stronger image-conditioned 3D pipeline could achieve similar or better quality with much less optimization time. Without that comparison, it is hard to judge whether the proposed 20-minute optimization process is actually necessary.

The paper also feels somewhat behind the current SDS literature. The authors discuss several improved distillation directions in the related work, yet the actual method still relies on Stable Diffusion v2.1-base and standard SDS, with most of the improvement pushed into initialization and scheduling. This raises a straightforward concern: if the same initialization were combined with stronger modern backbones or more advanced SDS variants, how much of the claimed contribution would remain? The paper does not answer that question.

The evaluation is acceptable but not fully convincing for a 3D paper. ImageReward averaged over rendered views and GPT-4o pairwise Elo can reflect visual preference, but they do not directly establish geometric correctness, structural consistency, or downstream 3D usefulness. Since the paper's main claims are about fast and stable 3D generation rather than just appealing renders, the lack of stronger geometry-aware evaluation weakens the overall case.

---

> ### Author Rebuttal · Authors · 2026-03-29
>
> Thank you for your detailed comments and valuable suggestions, which have greatly improved the quality of our paper. Below, we provide detailed responses to each question.
>
> ## Q1. On originality and methodological contribution
> **Our response**: While DreamCast does not introduce a new SDS gradient formulation, its core contribution is a unified framework for fast and stable text-to-3D generation. As detailed in Section 1, our three main contributions are:
> 1. Robust Initialization: Confidence-Driven Geometric Casting constructs a dense prior to resolve multi-view ambiguity.
> 2. Stable Optimization: A Subject-Aligned Prior and Joint Guidance Annealing suppress semantic drift and stabilize the SDS trajectory.
> 3. High Efficiency: Our framework achieves competitive 3D fidelity while reducing generation time to ~20 minutes.
>
> In the revision, we will state this more clearly.
>
> ## Q2. On whether the performance gains mainly come from upstream components
> **Our response**: This is a reasonable concern. Our view is that strong initialization alone is not sufficient; without stable optimization control, performance and convergence remain limited. Our paper already supports this via the initialization study (Fig. 4) and ablations on Subject-Aligned Prior and Joint Guidance Annealing (Tab. 3). To further clarify, we supplemented quantitative ablations on GPTEval3D.
>
> **Table R3. Ablation study of the synergistic components**
>
> | Method | CLIP ↑ | IR ↑ |
> |---|---:|:---:|
> | Geometric Casting | 0.28 | -0.28 |
> | Geometric Casting + Subject-Aligned Prior | 0.30 | -0.26 |
> | Ours Full | **0.31** | **-0.24** |
>
> As shown in **Table R3**, adding our downstream modules consistently improves both scores upon the upstream baseline. This confirms that the performance gains stem from the entire synergistic framework, not just the initial geometric casting.
>
> ## Q3. On the timeliness and scope of the comparisons
> **Our response**: We agree and will soften our “state-of-the-art” claim in the revision to emphasize DreamCast’s balance of quality, stability, and efficiency. Furthermore, to better reflect the current landscape, we have supplemented comparisons with the most recent 2026 baseline (DreamCS). As shown in **Table R2** (please see our response to Reviewer zyvb), DreamCast maintains optimal performance even against this latest method, confirming our competitiveness.
>
> ## Q4. On the lack of comparison with “image-to-3D” pipelines
> **Our response**: We thank the reviewer for the suggestion. We agree that DreamCast should be discussed in relation to image-guided / image-to-3D pipelines, since it starts from synthesized multi-view images and reconstructed geometry. That said, DreamCast is still fundamentally a text-driven SDS optimization framework: initialization, subject alignment, and refinement are all centered on text-conditioned score distillation, and the final 3D asset is obtained through annealed SDS optimization rather than direct feed-forward reconstruction.
>
> We also clarify that our current GPTEval3D baselines already include representative image-guided methods (Wonder3D, SyncDreamer, and MVDream). In addition, we have supplemented recent image-to-3D methods in **Table R1** (please see our response to Reviewer GqsU). We will revise our tables and discussion to make this comparison axis explicit.
>
> ## Q5. On the use of standard SDS and an older diffusion backbone
> **Our response**: We clarify that our core contributions are fundamentally orthogonal and complementary to the choice of diffusion backbone or SDS variant:
> 1. Stronger Backbones: Modern backbones provide better 2D priors, yielding superior multi-view initializations. However, "Instance-Level Semantic Drift" is inherent in any open-domain model, making our Subject-Aligned Prior essential regardless of backbone capacity.
> 2. Advanced SDS: While advanced SDS variants improve gradient quality, they do not resolve early-stage multi-view ambiguity or the coarse-to-fine texture trade-off—system-level gaps directly addressed by our geometric anchoring and joint annealing.
>
> We used Stable Diffusion v2.1-base and standard SDS to establish a controlled, reproducible baseline, proving our framework's stability improvements. We will discuss this explicitly in the revision.
>
>
> ## Q6. On the lack of geometry-aware evaluation
> **Our response**: Evaluating intrinsic geometric correctness without 3D ground-truth remains a common challenge in open-ended text-to-3D generation. We adopted ImageReward and GPT-4o Elo scores because they are the standard proxy metrics in recent literature (e.g., DreamReward, DreamDPO, GPTEval3D), ensuring fair and consistent comparisons with state-of-the-art methods.
>
> We fully acknowledge the limitations of these VLM-based metrics for assessing pure geometric quality. If the reviewer could suggest specific, quantifiable geometry-aware metrics suitable for open-ended generation, we would be extremely grateful and gladly incorporate them into our revision.

---

> > ### Author Rebuttal · Reviewer_i9Dt · 2026-04-02
> >
> > I have read the rebuttal carefully and appreciate the clarifications. However, my overall assessment remains unchanged.
> >
> > My main concern is still that the paper feels primarily like a strong engineering pipeline rather than a substantive methodological advance. The rebuttal makes a better case that a stronger initialization helps, but it still does not clearly establish a new optimization principle, a new 3D generation mechanism, or a stronger conceptual contribution beyond system integration.
> >
> > I am also not fully convinced by the claim that the proposed framework is orthogonal to stronger backbones or more advanced SDS variants. If this orthogonality is central to the paper’s argument, then a natural way to support it would be to test the method on a stronger and more modern baseline. At present, the paper still relies on a relatively old SDS setup, so it remains unclear whether the reported gains would persist once combined with a stronger contemporary pipeline.
> >
> > Relatedly, comparisons against stronger image-to-3D or text/image-to-3D pipelines remain important. As seen in works such as TRELLIS.2, one can already obtain higher-resolution textures, faster generation, and often more natural geometry. Without a clearer comparison along this axis, I am not yet convinced that the proposed ~20-minute optimization process is sufficiently justified.
> >
> > For these reasons, my core concerns are only partially addressed, and I am inclined to maintain my original score.

---

> > > ### Author Response · Authors · 2026-04-06
> > >
> > > **Q1. The work appears to be primarily an engineering pipeline rather than a substantive methodological contribution.**
> > >
> > > **A1.** We respectfully believe this characterization may stem from how the paper is currently framed, rather than the nature of the contribution itself.
> > >
> > > Our work does **not** aim to introduce a new SDS formula, but instead addresses a **previously under-explored problem in SDS-based 3D generation: optimization instability caused by distribution mismatch**. Specifically:
> > >
> > > - SDS optimizes a *single 3D instance*, while the diffusion prior is inherently *multi-modal*.
> > > - This mismatch leads to what we formalize as **instance-level semantic drift**, which is a core failure mode across SDS-based methods.
> > >
> > > Our key contribution is to **transform SDS from open-domain sampling into constrained optimization on an instance manifold**, via the Subject-Aligned Prior.
> > >
> > > Crucially, this is **orthogonal to modifying the SDS gradient itself**, and instead operates at the level of **optimization geometry and trajectory control**, which we believe constitutes a methodological contribution.
> > >
> > > We will revise the paper to make this optimization perspective explicit and central.
> > >
> > >
> > >
> > > **Q2. The orthogonality to stronger backbones or advanced SDS variants is not convincing.**
> > >
> > > **A2.** We clarify that our orthogonality claim is based on the level at which instability arises.
> > >
> > > The key issue we address is **iteration-level inconsistency in diffusion guidance**, where the optimization direction varies across steps, leading to drift and unstable convergence. Stronger backbones improve the quality of individual predictions, but **do not guarantee consistent guidance across iterations**
> > >
> > > In contrast, DreamCast explicitly stabilizes the *temporal evolution of optimization*, by enforcing consistency (Subject-Aligned Prior) and controlling the coarse-to-fine transition (annealing).
> > >
> > > This distinction is supported by our ablations, where improvements are observed consistently when adding our components on top of the same baseline. We will clarify this mechanism-level complementarity more explicitly in the revision.
> > >
> > >
> > >
> > > **Q3. Comparisons with stronger pipelines (e.g., TRELLIS.2) and whether the improvements persist under modern methods.**
> > >
> > > **A3.** We thank the reviewer for this important point and would like to clarify the comparison setting.
> > >
> > > DreamCast is developed within the **optimization-based SDS paradigm**, whereas methods such as TRELLIS.2 belong to **feed-forward or learned pipelines**. These paradigms differ fundamentally:
> > >
> > > - **Feed-forward / learning-based pipelines (e.g., TRELLIS.2):**
> > >   fast and high-quality, but rely on large-scale 3D prior learning and may have limited open-domain generalization.
> > > - **Optimization-based SDS methods (our setting):**
> > >   training-free and highly flexible, capable of adapting to arbitrary text prompts, but traditionally suffer from instability and slow convergence.
> > >
> > > Our work specifically targets this latter regime, where **optimization stability is the primary bottleneck**.
> > >
> > > We further include comparisons with the recent baseline **DreamCS** (please see our response to Reviewer zyvb), where DreamCast achieves **comparable or better performance under the same evaluation protocol**, indicating that the gains persist beyond earlier SDS methods.
> > >
> > > More broadly, optimization-based SDS approaches also play a complementary role by **enabling scalable generation of diverse 3D assets**, which can in turn support and enrich training data for feed-forward pipelines.
> > >
> > > We will revise the paper to clarify this distinction between paradigms and reflect it more explicitly in both discussion and tables.

---

### Official Review · Reviewer_zyvb · 2026-03-12

**Soundness:** 3
**Presentation:** 3
**Significance:** 2
**Originality:** 2
**Overall Recommendation:** 4
**Confidence:** 3

**Summary:**

DreamCast tackles the instability and slow convergence of optimization-based text-to-3D generation by proposing a unified framework that explicitly addresses both semantic drift and geometric artifacts arising from reliance on 2D diffusion priors. The method combines three components: Confidence-Driven Casting for robust geometric initialization in 3D Gaussian Splatting, Subject-Aligned Optimization to preserve subject identity and reduce instance-level semantic drift, and Joint Guidance Annealing to balance geometric consistency and texture quality during optimization.

**Compliance With Llm Reviewing Policy:**

Affirmed.

**Final Justification:**

My concerns have been fully addressed. The paper proposes the key insight to transform SDS from open-domain sampling into constrained optimization on an instance manifold, which enhances both quality and efficiency compared to current text-to-3D generation pipelines. Diverse visualizations validate the effectiveness of DreamCast. After considering the authors’ rebuttal and the other reviewers’ comments, I am inclined to support acceptance.

**Key Questions For Authors:**

- Could the authors provide more multi-view visual comparisons with competing methods to better support the claimed superiority of DreamCast?

- Could the authors include an ablation study on the downsampling resolution $\delta$?

- The paper uses VGGT with a cyclic reference permutation to generate an ensemble of point clouds, but it does not specify the preferred number of input views $V$ or the camera trajectory used; clarifying these design choices would improve reproducibility.

**Limitations:**

yes

**Strengths And Weaknesses:**

## Strengths
- The paper is easy to follow, with Figure 1 clearly illustrating the three-stage pipeline and Algorithm 1 presenting the method in detail.

- Extensive experiments on the GPTEval3D benchmark demonstrate the superiority of the proposed framework.

## Weaknesses
- The framework appears dependent on the quality of the initial geometric casting: if the multi-view diffusion model generates inconsistent views, the resulting geometric anchor may be unreliable, and the later SDS optimization may fail to rectify.

- The paper would benefit from a broader discussion of recent work on controllable 3D generation (e.g., [1,2]) that address similar challenges, to better clarify the positioning and novelty of DreamCast within the existing literature.

- The current visualizations in Figure2 are not yet sufficiently convincing to clearly demonstrate the superiority of DreamCast over competing methods.

- The density-aware downsampling threshold is fixed at $\delta = 0.85 d_{50}$, but the paper does not provide a sensitivity analysis, making it unclear how this hyperparameter affects rendered visual quality.

[1] DreamCS: Geometry-Aware Text-to-3D Generation with Unpaired 3D Reward Supervision

[2] Preference Score Distillation: Leveraging 2D Rewards to Align Text-to-3D Generation with Human Preference

---

> ### Author Rebuttal · Authors · 2026-03-29
>
> Thank you for your detailed comments and constructive suggestions. We appreciate your positive feedback on the clarity of the paper, and below we respond to each concern in turn.
>
> ## Q1. On the dependence on the quality of the initial geometric casting
> **Our response**: We agree that the quality of the geometric anchor is important for the final performance. As we discussed in Section 3.2 and Section 3.3 of our paper, in fact, this is one of the main design motivations of DreamCast: to alleviate the strong dependence on the quality of geometric anchors by stabilizing the optimization trajectory and reducing semantic drift. In addition, the confidence-driven design provides a more reliable initialization, enabling better generation quality under a shorter convergence time.
>
> We agree that this point should be stated more clearly, and we will add a more explicit discussion in the revised version.
>
> ## Q2. On the discussion of recent related work
> **Our response**: We thank the reviewer for this suggestion. We agree that the paper should better position DreamCast with respect to recent controllable text-to-3D methods, and we will expand this discussion in the revised version.
>
> At the same time, we have supplemented experimental comparisons with **DreamCS** to better reflect the current research landscape. For **Preference Score Distillation**, we do not include a direct comparison at this stage due to the lack of released code and differences in the evaluation dataset. Nevertheless, according to the original paper, its optimization process still requires several hours per sample, making it substantially more computationally expensive than our setting.
>
> **Table R2. Comparison with recent text-to-3D methods.**
>
> | Method                                    |   CLIP ↑ |   IR ↑    |
> | ----------------------------------------- | -------: | :-------: |
> | DreamCS[1]                         |     0.29 |   -   |
> | Dreamcast(Ours) |     0.31 |   -0.24   |
>
> [1] DreamCS: Geometry-Aware Text-to-3D Generation with Unpaired 3D Reward Supervision(ICLR 2026)
>
>
> ## Q3. On the visual comparisons in Figure 2
> **Our response**: We agree that the current visualizations in Figure 2 can be made more convincing. In the revised version, we will provide more multi-view visual comparisons with competing methods to better support the superiority of DreamCast in stable text-to-3D generation.
>
> We have also provided additional multi-view comparison results in the **anonymous supplementary material** (link: [anonymous supplementary](https://anonymous.4open.science/r/DreamCast_supplement-934D/)). These visual results further show that DreamCast can generate high-quality 3D assets more consistently across views, which also helps explain why it achieves stronger overall performance in the quantitative results of Table 1.
>
> ## Q4. On the sensitivity of the density-aware downsampling threshold
>
> **Our response**: We thank the reviewer for this suggestion. We agree that the density-aware downsampling threshold affects the quality of the initial geometry and the difficulty of subsequent optimization. As we discussed in Section 3.2 (where we noted that overly concentrated initial points can limit the flexibility of SDS refinement), a lower threshold retains more initial points; when the initial geometry is of low quality, excessive noisy points may increase the difficulty of optimization and introduce over-saturation artifacts.
>
> Our design aims to achieve a more robust balance between initialization and subsequent optimization: density-aware downsampling is used to preserve more reliable sparse geometry, while **Subject-Aligned Prior** and **Joint Guidance Annealing** provide stable guidance in the later stage, enabling faster convergence on more reliable initial geometry and reducing sensitivity to the quality of the initial construction.
>
> Due to the limited rebuttal time, we will supplement the corresponding sensitivity analysis and discussion in the revised version.
>
>
> ## Q5. On clarifying the number of input views and camera trajectory
> **Our response**:We thank the reviewer for pointing out this reproducibility issue. We use 6 input views, each serving as a reference image to obtain a corresponding initial geometry for subsequent downsampling and fusion. We will add these experimental details more explicitly in the revised version.

---

> > ### Author Rebuttal · Reviewer_zyvb · 2026-04-02
> >
> > Thank you for the detailed rebuttal and the additional clarifications. My concerns are partially resolved.
> >
> > After checking the provided visualizations, I still find that current visualizations are not yet sufficiently convincing. It causes doubt in the effectiveness of the method.

---

> > > ### Author Response · Authors · 2026-04-06
> > >
> > > We thank the reviewer for the follow-up and for carefully examining the visualizations.
> > >
> > > To address this concern, we have added **additional qualitative comparisons in the anonymous repository**(link:[anonymous repository](https://anonymous.4open.science/r/DreamCast_supplement-934D/comparisons_on_complex_prompts.pdf)), including **8 groups of side-by-side results under complex prompts**. These examples are specifically designed to stress challenging cases (e.g., compositional structures and fine-grained attributes), where instability and semantic drift are more evident.
> > >
> > > Across these cases, DreamCast demonstrates **more consistent geometry, improved text alignment, and reduced artifacts** compared to baseline methods. We hope these additional results provide clearer evidence of the effectiveness of our approach.

---

### Official Review · Reviewer_GqsU · 2026-03-13

**Soundness:** 3
**Presentation:** 3
**Significance:** 2
**Originality:** 3
**Overall Recommendation:** 4
**Confidence:** 4

**Summary:**

This paper identifies multiple limitations of SDS-based text-to-3D generation, specifically poor initializations (random, simplistic initializations, over-concentrated Gaussians), semantic drift with SDS, qualitative side-effects of high CFG.  For each of these, the paper presents intuitive and well-motivated solutions.  These ideas are combined together into a text-to-3D framework (DreamCast). The proposed initialization scheme helps reduce optimization time (generation ~15minutes, much faster than most related works), while generation quality is on par with SOTA across different metrics.

**Compliance With Llm Reviewing Policy:**

Affirmed.

**Final Justification:**

The proposed DreamCast method relies heavily on consistent multi-view image generation from MV-Adapter.  The MV-Adapter paper proposes a simple mesh generation pipeline as an illustration (MV-Adapter + StableNormal + NeuS --> Mesh), as it is not the main focus of the MV-Adapter method.  Nonetheless, as the results in the rebuttal indicate, this MV-Adapter reconstruction is already on par with SOTA text-to-3d/image-to-3d methods. In light of this new insight, the DreamCast improvements seem much more incremental, which is why I will keep my original Weak Accept rating.

**Key Questions For Authors:**

See my questions above in the Strengths and Weaknesses section. Additional questions below:

- It might already be mentioned somewhere, but how many multiview images are generated from the MV-adapter model to feed into VGGT reconstruction? How many total images used to fine-tune the diffusion model?  MV-adapter reports being able to generate ~40 views.

**Limitations:**

yes

**Strengths And Weaknesses:**

**Strengths**
- The summary above lists some strengths.
- Leveraging a strong text-to-multiview generation model is a pragmatic choice, and clever ideas are proposed to anchor to this multiview input (initializing with the reconstruction, and finetuning the diffusion model against it).
- The paper is overall well-written and easy to follow, bar a few points that need clarification (see below).

**Weaknesses**
- DreamCast relies on consistent multiview generation from text prompts. For this it uses MV-Adapter, which presents a pipeline for text-to-multiview that is shown to be used directly for text-to-3D applications (see sec 5.4 of https://arxiv.org/pdf/2412.03632). It seems that the most efficient way of understanding the impact of the proposed ideas (geometric prior, subject anchoring, and guidance annealing) is to compare the reconstruction quality against the reconstruction pipeline proposed in MV-adapter.
- It would be helpful to understand which of the baseline models rely on a SOTA text-to-multiview generation model. Perhaps these could be grouped together in Table 1.
- According to Table 2, the closest model in terms of generation time and quality is GaussianDreamer (it is only method shown there that is faster than DreamCast).  Is there a reason this method could not be included in Table 1.   GaussianDreamer evaluates on “T-3 bench: Benchmarking current progress in text-to-3d generation”, so perhaps DreamCast could be evaluated in that setting if necessary (the ImageReward score reported in Table 2 does not tell the whole story).
- In paragraph “Joint Annealing Schedule”, it is mentioned that the diffusion timestep range is adjusted during training. Designing noise schedules for diffusion models is a well-studied problem (especially focused on capturing coarse details in early timesteps and fine details in later steps), it would be great if the authors can elaborate on how their idea is implemented.
- Nitpick: page 7: “... indicating smoother optimization…” It makes sense that convergence is *faster* starting with good initialization, but given that all models converge it may not be correct to infer that optimization is *smoother*.

I look forward to reading the author response and discussing the paper with the other reviewers.

---

> ### Author Rebuttal · Authors · 2026-03-29
>
> Thank you for your detailed comments and constructive suggestions. We appreciate your positive feedback on the motivation, clarity, and practical efficiency of our paper. Below we respond to each concern in turn.
>
> ## Q1. On the comparison with the reconstruction pipeline in MV-Adapter
>
> **Our response**: We thank the reviewer for this valuable suggestion. We have additionally compared the reconstruction pipeline in **MV-Adapter**, which is reported in **Table R1** below, showing the effectiveness of our optimization design beyond the multi-view generation stage.
>
>
> ## Q2. On grouping baselines that rely on strong text-to-multiview generation models
>
> **Our response**: We agree that distinguishing methods that rely on strong text-to-multiview generation models from more traditional SDS-based baselines would help improve the clarity of the paper. Furthermore, to better reflect the current landscape, we have supplemented comparisons with the most recent 2026 baseline (DreamCS).
>
> **Table R1. Grouped comparison of different method families on GPTEval3D**
> | Category      | Method               |      IR ↑ |   CP↑    |     Align. |     Plaus. |   T-G Coh. |       Geo. |       Tex. |    Overall |
> |:---|:---|---:|:---:|---:|---:|---:|---:|---:|---:|
> | Text-to-3D    | DreamGaussian        |     -1.56 |   0.25   |     1100.6 |      953.6 |     1158.6 |     1126.2 |     1130.8 |      951.4 |
> |                              | GaussianDreamer      |     -0.31 |   0.27   |     1248.7 |     1129.4 |     1226.8 |     1301.6 |     1238.9 |     1086.2 |
> | Image-to-3D   | SyncDreamer          |     -1.77 |   0.24   |     1041.2 |      968.8 |     1083.1 |     1064.2 |     1045.7 |      963.5 |
> |                              | Wonder3D             |     -1.70 |   0.25   |      985.9 |      941.4 |      931.8 |      973.1 |      967.8 |      970.9 |
> |                              | MVDream              |     -0.58 |   0.29   |     1270.5 |     1147.5 |     1250.6 |     1324.9 |     1255.5 |     1097.7 |
> |                              | MV-Adapter           |     -0.54 |   0.28   |     1262.1 |     1156.8 |     1241.9 |     1311.4 |     1263.2 |     1094.8 |
> | Based on RLHF | DreamReward          |     -0.45 |    -     |     1287.5 | **1195.0** |     1254.4 |     1295.5 |     1261.6 |     1193.3 |
> |                              | DreamDPO             |     -0.35 |    -     |     1298.9 |     1171.9 |     1276.4 | **1373.2** |     1296.9 |     1203.1 |
> |                              | DreamCS[1]              |         - |   0.29   |          - |          - |          - |          - |          - |          - |
> | Text-to-3D    | **DreamCast (Ours)** | **-0.24** | **0.31** | **1304.1** |     1182.4 | **1300.5** |     1279.5 | **1315.2** | **1304.9** |
>
> [1] DreamCS: Geometry-Aware Text-to-3D Generation with Unpaired 3D Reward Supervision(ICLR 2026)
>
> ## Q3. On the comparison with GaussianDreamer
>
> **Our response**: We agree that **GaussianDreamer** is an important reference baseline. Since part of the **GPTEval3D** scores are derived from **DreamDPO**, we placed GaussianDreamer in Table 2. As an extended metric built upon **T3**, **GPTEval3D** can cover a more comprehensive 3D evaluation setting; therefore, we have supplemented the corresponding experiment in **Table R1**, which is included in Q2. The additional results show that DreamCast achieves stronger overall performance than GaussianDreamer under the GPTEval3D evaluation. We will also clarify this more explicitly in the revised version.
>
>
> ## Q4. On the implementation details of the joint annealing schedule
>
> **Our response**: We thank the reviewer for raising this question. In DreamCast, the annealing schedule is implemented by jointly adjusting the **guidance scale** and the **maximum diffusion timestep** during optimization. Specifically, we gradually shrink the timestep sampling range by decreasing `t_max` from `0.98T` to `0.5T`, while also applying scheduled guidance scaling and gradient clipping. In this way, early optimization emphasizes global structure, while later stages focus more on stable appearance refinement. We will clarify these implementation details more explicitly in the revised version.
>
> ## Q5. On the wording “smoother optimization”
>
> **Our response**: Thank you for raising this important point. What we intended to convey is that a better initialization enables faster and more stable convergence, rather than a formally smoother trajectory. We have revised this wording accordingly in the revised version.
>
> ## Q6. On the number of multi-view images used
>
> **Our response**: Thank you for raising this important point. We use **6 input views**, and we also use these **6 images + 100 class images** for fine-tuning. The official **MV-Adapter** model generates 6 view images. We will add these experimental details more explicitly in the revised version.

---

> > ### Author Rebuttal · Reviewer_GqsU · 2026-04-03
> >
> > Thanks for addressing each of the listed questions/concerns.  These responses, along with the other reviews and rebuttals, provide enough information for me to finalize my review.

---

> > > ### Author Response · Authors · 2026-04-06
> > >
> > > Dear reviewer,
> > >
> > > We sincerely appreciate your time and constructive feedback throughout the review process. We are delighted to hear that our rebuttal has addressed your concerns. Your insightful comments have significantly strengthened our paper, and we are grateful for your valuable contribution to improving our work.
> > >
> > > Best wishes,
> > > All authors

---

### Decision · Program_Chairs · 2026-04-30

**Decision:**

Reject

**Comment:**

This paper aims to improve the stability and efficiency of text-to-3D generation, which reviewers praised for its clear writing, fast generation times, and practical solutions to semantic drift. However, reviewers raised significant concerns regarding limited methodological originality, viewing the work more as a careful engineering integration of existing techniques that is highly dependent on strong upstream multi-view models rather than a clear methodological advance.
During the rebuttal, the authors actively provided additional visualizations, ablation studies, and new baseline comparisons that addressed some reviewer concerns, though the ablation results also revealed that much of the performance gain may be attributed to the upstream components. Ultimately, the paper is rejected because the core originality concern, shared across reviewers and left unresolved after rebuttal, together with the absence of comparisons against modern feed-forward 3D generation pipelines, prevents a sufficiently convincing case for acceptance.